# Mutation-Driven Follow the Regularized Leader for Last-Iterate Convergence in Zero-Sum Games

**Kenshi Abe**[1]  **Mitsuki Sakamoto**[2]  **Atsushi Iwasaki**[2]

[1]CyberAgent, Inc.
[2]University of Electro-Communications

## Abstract

In this study, we consider a variant of the Follow the Regularized Leader (FTRL) dynamics in two-player zero-sum games. FTRL is guaranteed to converge to a Nash equilibrium when time-averaging the strategies, while a lot of variants suffer from the issue of limit cycling behavior, i.e., lack the last-iterate convergence guarantee. To this end, we propose mutant FTRL (M-FTRL), an algorithm that introduces mutation for the perturbation of action probabilities. We then investigate the continuous-time dynamics of M-FTRL and provide the strong convergence guarantees toward stationary points that approximate Nash equilibria under full-information feedback. Furthermore, our simulation demonstrates that M-FTRL can enjoy faster convergence rates than FTRL and optimistic FTRL under full-information feedback and surprisingly exhibits clear convergence under bandit feedback.

## 1 INTRODUCTION

Our study focuses on the problem of learning an equilibrium in two-player zero-sum games. In order to find an equilibrium in two-player zero-sum games, we need to solve a minimax optimization (or saddle-point optimization) in the form of $\min_x \max_y f(x, y)$. Motivated by advances of multi-agent reinforcement learning [Busoniu et al., 2008] and Generative Adversarial Networks (GANs) [Goodfellow et al., 2014], the development of algorithms that efficiently approximate the solution of the minimax optimization is attracting considerable interest [Blum and Monsour, 2007, Daskalakis et al., 2018].

There are a lot of studies focusing on developing no-regret learning algorithms where the iterate-average strategy profile converges to a Nash equilibrium of two-player zero-sum games [Banerjee and Peng, 2005, Zinkevich et al., 2007, Daskalakis et al., 2011]. However, well-known no-regret learning algorithms such as Follow the Regularized Leader (FTRL) are shown to cycle and fail to converge without time-averaging [Mertikopoulos et al., 2018, Bailey and Piliouras, 2018]. In recent years, several studies have developed and analyzed algorithms whose trajectory of updated strategies directly converges to an equilibrium without forming a cycle, such as optimistic FTRL (O-FTRL) [Daskalakis et al., 2018, Daskalakis and Panageas, 2019, Mertikopoulos et al., 2019, Wei et al., 2021, Lei et al., 2021]. This convergence property is known as *last-iterate convergence*. However, establishing the explicit convergence rates of optimistic multiplicative weights update, which is tantamount to O-FTRL with entropy regularization, requires that the equilibrium in underlying games must be unique [Daskalakis and Panageas, 2019, Wei et al., 2021].

In this study, as an alternative, we propose mutant FTRL[1] (M-FTRL), an algorithm that introduces mutation for the perturbation of action probabilities. We first identify the discrete-time version of the M-FTRL dynamics and then modify it to the continuous-time version to provide the theoretical analysis. We prove the followings: 1) M-FTRL dynamics induced by the entropy regularizer is equivalent to replicator-mutator dynamics (RMD) [Hofbauer et al., 2009, Zagorsky et al., 2013, Bauer et al., 2019]; 2) for general regularization functions, the strategy trajectory of M-FTRL converges to a stationary point of the RMD; 3) the trajectory of M-FTRL with the entropy regularizer converges to an approximate Nash equilibrium at an exponentially fast rate. To the best of our knowledge, we are the first to provide the convergence result for RMD in two-player zero-sum games.

Furthermore, our simulation demonstrates that M-FTRL can enjoy faster convergence rates than FTRL and optimistic FTRL under full-information feedback, i.e., M-FTRL converges to a stationary point, which approximates a Nash

---

[1]An implementation of our method is available at https://github.com/CyberAgentAILab/mutant-ftrl.

*Accepted for the 38th Conference on Uncertainty in Artificial Intelligence* (UAI 2022).

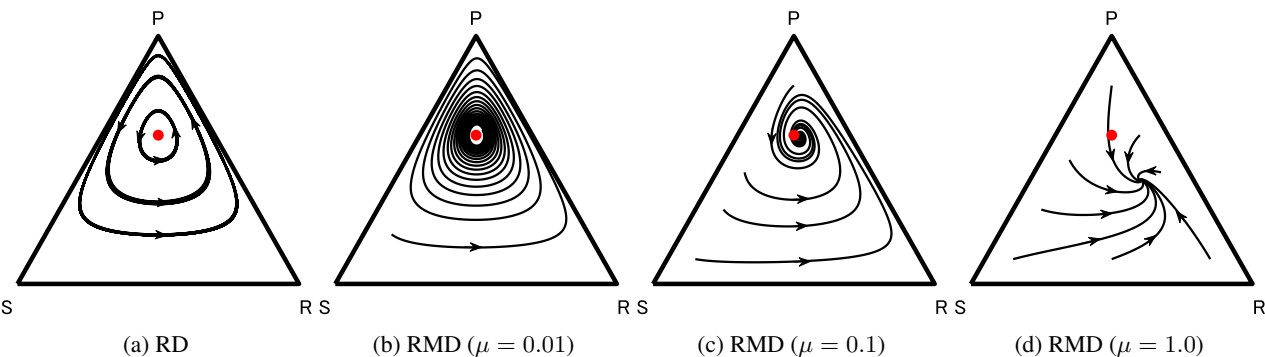

Figure 1: Learning dynamics of RD and RMD in biased Rock-Paper-Scissors. The red dot represents the Nash equilibrium point of the game.

equilibrium, faster. It also exhibits clear convergence under partial-information or bandit feedback, where each player takes the feedback about the payoffs from his or her chosen actions. We empirically observe the last-iterate convergence behavior in the M-FTRL dynamics, as well as under full-information feedback, while neither FTRL nor O-FTRL reveals such behavior. This is surprising because it is an open question if a last-iterate convergence guarantee is provided under bandit feedback.

## 2 RELATED LITERATURE

**Average-iterate convergence** There are a lot of previous studies focusing on developing no-regret learning algorithms that enjoy average-iterate convergence in two-player zero-sum games [Cesa-Bianchi and Lugosi, 2006, Zinkevich et al., 2007, Hofbauer et al., 2009, Syrgkanis et al., 2015]. FTRL is one of the most widely studied no-regret learning algorithm and has been shown to be convergent if the equilibrium is deterministic or strict [Mertikopoulos et al., 2018, Giannou et al., 2021]. If the equilibrium strategy is a mixed strategy with full support, FTRL's trajectory can be recurrent [Mertikopoulos et al., 2018]. For extensive-form games, counterfactual regret minimization [Zinkevich et al., 2007] and its variants have been developed as a no-regret learning algorithm [Gibson et al., 2012, Tammelin, 2014, Lanctot et al., 2017, Schmid et al., 2019, Brown and Sandholm, 2019, Davis et al., 2020]. However, most of these algorithms have not been proven that the last-iterate strategy converges.

**Last-iterate convergence** In recent years, various algorithms using an optimistic online learning framework [Rakhlin and Sridharan, 2013a,b] have been proposed for last-iterate convergence in minimax optimization. Optimistic gradient descent ascent [Daskalakis et al., 2018, Mertikopoulos et al., 2019, Wei et al., 2021] and optimistic multiplicative weights update [Daskalakis and Panageas, 2019, Wei et al., 2021, Lei et al., 2021] are the variants of O-FTRL,

and they have been shown to enjoy the last-iterate convergence guarantee in constrained and unconstrained saddle optimization problems. Furthermore, Nguyen et al. [2021] have proposed the no-regret learning algorithm, which exhibits the last-iterate convergence in asymmetric repeated games. In contrast to their optimistic modification of FTRL, which boosts updates for expected utitilities, our method is motivated by replicator-mutator dynamics and provides an alternative way to enjoy the last-iterate convergence guarantee.

**Replicator-mutator dynamics** Evolutionary game theory has been strongly related to learning dynamics. In fact, it is well-known that cross learning converges to the replicator dynamics (RD) in the continuous-time limit [Börgers and Sarin, 1997, Bloembergen et al., 2015], similarly to FTRL. On the other hand, RMD [Hofbauer and Sigmund, 1998] has been overlooked in the context of learning. Introducing mutation empirically makes numerical errors in computation small [Zagorsky et al., 2013]. However, it makes difficult to analyze the properties. Some notable exceptions report that mutation stabilizes the dynamics [Bomze and Burger, 1995, Bauer et al., 2019]. Let $\pi^\mu$ be an interior stationary point of RMD with mutation rate $\mu$, then $\pi^\mu$ is $\varepsilon$-Nash equilibrium of the underlying game for $\varepsilon = \mu$ [Bauer et al., 2019]. Also, evolutionary game dynamics such as RD typically exhibits continua of stationary points and is unlikely to converge to a unique, stable stationary point. Mutation dissolves continua of neutrally stable equilibria into isolated, asymptotically stable ones [Bomze and Burger, 1995].

## 3 PRELIMINARIES

### 3.1 TWO-PLAYER ZERO-SUM NORMAL-FORM GAME

A two-player normal-form game is defined by utility functions $u_i \in [-u_{\max}, u_{\max}]^{A_1 \times A_2}$, where $A_i$ is the finite action space for player $i \in \{1, 2\}$. In a two-player zero-sum

normal-form game, $u_i$ satisfies $u_1(a_1, a_2) + u_2(a_1, a_2) = 0$ for all $a_1 \in A_1$ and $a_2 \in A_2$. In this game, each player $i$ selects action $a_i \in A_i$ simultaneously. Then, player $i$ receives utility $u_i(a_1, a_2)$. Let us denote $\pi_i \in \Delta(A_i)$ as a *mixed strategy* for player $i$, where $\Delta(A_i) := \{p \in [0,1]^{|A_i|} \mid \sum_{a_i \in A_i} p(a_i) = 1\}$ represents the probability simplex on $A_i$. We define a *strategy profile* as $\pi = (\pi_1, \pi_2)$. For a given strategy profile $\pi$, the expected utility for player $i$ is given by $v_i^\pi = \mathbb{E}_{a \sim \pi}[u_i(a_1, a_2)]$. We further define the conditional expected utility of taking action $a_i \in A_i$ as $q_i^\pi(a_i) = \mathbb{E}_{a_{-i} \sim \pi_{-i}}[u_i(a_i, a_{-i}) | a_i]$, where $-i$ represents the opponent of player $i$. Finally, we denote the conditional expected utility vector as $q_i^\pi = (q_i^\pi(a_i))_{a_i \in A_i}$.

## 3.2 NASH EQUILIBRIUM AND EXPLOITABILITY

A common solution concept for two-player games is a *Nash equilibrium* [Nash, 1951], where no player cannot improve his/her expected utility by deviating from his/her specified strategy. In two-player zero-sum normal-form games, a Nash equilibrium $\pi^* = (\pi_1^*, \pi_2^*)$ ensures the following condition: $\forall \pi_1 \in \Delta(A_1), \forall \pi_2 \in \Delta(A_2)$,

$$v_1^{\pi_1^*, \pi_2} \geq v_1^{\pi_1^*, \pi_2^*} \geq v_1^{\pi_1, \pi_2^*}.$$

An $\epsilon$-*Nash equilibrium* $(\pi_1, \pi_2)$ is an approximation of a Nash equilibrium, which satisfies the following inequality:

$$\max_{\tilde{\pi}_1 \in \Delta(A_1)} v_1^{\tilde{\pi}_1, \pi_2} + \max_{\tilde{\pi}_2 \in \Delta(A_2)} v_2^{\pi_1, \tilde{\pi}_2} \leq \epsilon.$$

Furthermore, we call $\operatorname{exploit}(\pi) := \max_{\tilde{\pi}_1 \in \Delta(A_1)} v_1^{\tilde{\pi}_1, \pi_2} + \max_{\tilde{\pi}_2 \in \Delta(A_2)} v_2^{\pi_1, \tilde{\pi}_2}$ as *exploitability* of a given strategy profile $\pi$. Exploitability is a metric for measuring how close $\pi$ is to a Nash equilibrium $\pi^*$ in two-player zero-sum games [Johanson et al., 2011, 2012, Lockhart et al., 2019, Timbers et al., 2020, Abe and Kaneko, 2021]. From the definition, a Nash equilibrium $\pi^*$ has the lowest exploitability of 0.

## 3.3 PROBLEM SETTING

In this study, we consider the setting where the game is played repeatedly for $T$ iterations. At each iteration $t \in [T]$, each player $i$ determines the (mixed) strategy $\pi_i^t \in \Delta(A_i)$ based on the past-observed feedback. Then, each player $i$ observes the new feedback. In this study, we focus on two feedback cases: *full-information feedback* and *bandit feedback*. At the end of the iteration $t$ under full-information feedback, player $i$ observes the conditional expected utility vector $(q_i^{\pi^t}(a_i))_{a_i \in A_i}$ as feedback. Under bandit feedback, each player $i$ chooses an action $a_i^t$ according to $\pi_i^t$. Then, each player observes the realized utility $u_i(a_1^t, a_2^t)$.

FTRL is a widely used learning algorithm in the repeated game setting. For player $i$, FTRL methods are defined with *regularization function* $\psi_i : \Delta(A_i) \to \mathbb{R}$, which is strictly convex and continuously differentiable on $\Delta(A_i)$. In FTRL, each player $i$ determines her strategy $\pi_i^t$ at iteration $t$ as follows:

$$\pi_i^t = \arg\max_{p \in \Delta(A_i)} \left\{ \eta \langle y_i^t, p \rangle - \psi_i(p) \right\},$$

$$y_i^t(a_i) = \sum_{s=1}^{t-1} q_i^{\pi^s}(a_i),$$

where $\eta > 0$ is the learning rate.

## 3.4 OTHER NOTATIONS

We denote the interior of the probability simplex $\Delta(A_i)$ by $\Delta^\circ(A_i) := \{p \in \Delta(A_i) \mid \forall a_i \in A_i, \ p(a_i) > 0\}$. For a strictly convex and continuously differentiable function $\psi$, the associated *Bregman divergence* is defined as $D_\psi(x, x') = \psi(x) - \psi(x') - \langle \nabla \psi(x'), x - x' \rangle$. The *Kullback-Leibler divergence*, which is the Bregman divergence with the entropy regularizer $\psi(x) = \sum_i x_i \ln x_i$, is denoted by $\mathrm{KL}(x, x') = \sum_i x_i \ln \frac{x_i}{x_i'}$. Besides, we define the sum of Bregman divergences and sum of Kullback-Leibler divergences as $D_\psi(\pi, \pi') = \sum_{i=1}^2 D_{\psi_i}(\pi_i, \pi_i')$ and $\mathrm{KL}(\pi, \pi') = \sum_{i=1}^2 \mathrm{KL}(\pi_i, \pi_i')$, respectively.

# 4 MUTANT FOLLOW THE REGULARIZED LEADER

In this section, we introduce *Mutant Follow the Regularized Leader* (M-FTRL), which is inspired by the RMD [Hofbauer and Sigmund, 1998, Zagorsky et al., 2013]. Let us see what happens in a biased version of the Rock-Paper-Scissors game, see Table 1. Figure 1 compares trajectories of RD and RMD with varying mutation parameters $\mu$ (see (RMD) for the differential equation of RMD). Note that $\mu$ represents the parameter that controls the strength of mutation. Figure 1a shows that the trajectories form a cycle and never converge to the Nash equilibrium because the game is intransitive. Note, however, that the time-averaged trajectory of FTRL converges to interior Nash equilibria in two-player zero-sum games [Hofbauer et al., 2009]. In contrast, Figures 1b and 1c exhibit a clear convergence to the unique stationary point, which is almost equivalent to the interior Nash equilibrium (the red dot) without taking the time average. As the mutation parameter increases to 1.0, although the stationary point becomes far from the Nash equilibrium, it is still asymptotically stable in Figure 1d. Thus, mutation is expected to ensure that the trajectory of a learning dynamics reaches an approximated equilibrium.

## 4.1 ALGORITHM

We propose a discrete-time version of the M-FTRL algorithm under two feedback cases: full-information feedback

**Algorithm 1** Mutant Follow the Regularized Leader with adaptive reference strategies for player $i$.

**Require:** Time horizon $T$, learning rate $\eta$, regularization function $\psi_i$, mutation parameter $\mu$, update frequency $N$, initial strategy $\pi_i^0$

1: $c_i \leftarrow \left( \frac{1}{|A_i|} \right)_{a_i \in A_i}$
2: $\tau \leftarrow 0$
3: Initialize $z_i^0$ so that $\pi_i^0 = \arg\max_{p \in \Delta(A_i)} \left\{ \langle z_i^0, p \rangle - \psi_i(p) \right\}$
4: **for** $t = 1, 2, \cdots, T$ **do**
5:    Compute strategy $\pi_i^t$ by
$$\pi_i^t = \arg\max_{p \in \Delta(A_i)} \left\{ \langle z_i^t, p \rangle - \psi_i(p) \right\}$$
6:    **for** $a \in A_i$ **do**
7:       $z_i^{t+1}(a) \leftarrow z_i^t(a) + \eta \left( q_i^{\pi^t}(a) + \frac{\mu}{\pi_i^t(a)}(c_i(a) - \pi_i^t(a)) \right)$
8:    **end for**
9:    $\tau \leftarrow \tau + 1$
10:    **if** $\tau = N$ **then**
11:       $c_i \leftarrow \pi_i^t$
12:       $\tau \leftarrow 0$
13:    **end if**
14: **end for**

and bandit feedback. First, we provide the strategy update rule under full-information feedback:

$$\pi_i^t = \arg\max_{p \in \Delta(A_i)} \left\{ \eta \left\langle \sum_{s=1}^{t-1} q_i^{\mu,s}, p \right\rangle - \psi_i(p) \right\}, \quad (1)$$

$$q_i^{\mu,s}(a_i) = q_i^{\pi^s}(a_i) + \frac{\mu}{\pi_i^s(a_i)}(c_i(a_i) - \pi_i^s(a_i)),$$

where $\eta > 0$ is the learning rate, $\mu > 0$ is the *mutation parameter*, and $c_i \in \Delta^\circ(A_i)$ is the *reference strategy*.

As shown in Figure 1b-1d, strategies $\pi_i^t$ updated by (1) would converge to the stationary point, which is different from the Nash equilibrium of the original game. The stationary point is a $2\mu$-Nash equilibrium of the original game, and the stationary point is not Nash equilibrium unless $(c_1, c_2)$ is a Nash equilibrium (see Theorem 5.4). Therefore, for convergence to a Nash equilibrium of the original game, we introduce a technique to adapt the reference strategy. That is, we copy probabilities from $\pi_i^t$ into $c_i$ every $N(\leq T)$ iterations. This technique is similar to the direct convergence method by [Perolat et al., 2021]. The pseudo-code of our algorithm with adaptive reference strategies is presented in Algorithm 1.

Under bandit feedback, each player $i$ needs to estimate $q_i^{\mu,t} = \left( q_i^{\pi^t}(a_i) + \frac{\mu}{\pi_i^t(a_i)}(c_i(a_i) - \pi_i^t(a_i)) \right)_{a_i \in A_i}$ from the realized utility $u_i(a_1^t, a_2^t)$. Similarly to [Wei and Luo,

2018, Ito, 2021], we construct the following estimator $\hat{q}_i^{\mu,t}$:

$$\hat{q}_i^{\mu,t}(a_i) = \frac{u_i(a_1^t, a_2^t)}{\pi_i^t(a_i^t)} \mathbb{1}[a_i = a_i^t] + \frac{\mu}{\pi_i^t(a_i)}(c_i(a_i) - \pi_i^t(a_i)). \quad (2)$$

It is easy to confirm that $\hat{q}_i^{\mu,t}$ is an unbiased estimator of $q_i^{\mu,t}$. Under bandit feedback, M-FTRL updates the strategy $\pi_i^t$ by the following update rule, which uses $\hat{q}_i^{\mu,t}$ instead of $q_i^{\mu,t}$ in (1):

$$\pi_i^t = \arg\max_{p \in \Delta(A_i)} \left\{ \eta \left\langle \sum_{s=1}^{t-1} \hat{q}_i^{\mu,s}, p \right\rangle - \psi_i(p) \right\}.$$

Note that M-FTRL does not require any information about the opponent's strategy $\pi_{-i}^t$ under bandit feedback.

## 5 THEORETICAL ANALYSIS

In this section, we provide the theoretical relationship between RMD and M-FTRL and the last-iterate convergence guarantee of M-FTRL. Instead of the discrete-time version of M-FTRL algorithm, we analyze the theoretical properties of the following continuous-time version of M-FTRL dynamics:

$$\pi_i^t = \arg\max_{p \in \Delta(A_i)} \left\{ \langle z_i^t, p \rangle - \psi_i(p) \right\}, \quad (3)$$

$$z_i^t(a_i) = \int_0^t \left( q_i^{\pi^s}(a_i) + \frac{\mu}{\pi_i^s(a_i)}(c_i(a_i) - \pi_i^s(a_i)) \right) ds.$$

First, we show that this dynamics is a generalization of RMD [Bauer et al., 2019]. That is, the dynamics of M-FTRL with the entropy regularizer $\psi_i(p) = \sum_{a_i \in A_i} p(a_i) \ln p(a_i)$ induces RMD:

**Theorem 5.1.** *The dynamics defined by (3) with the entropy regularizer $\psi_i(p) = \sum_{a_i \in A_i} p(a_i) \ln p(a_i)$ is equivalent to replicator-mutator dynamics:*

$$\frac{d}{dt}\pi_i^t(a_i) = \pi_i^t(a_i)\left( q_i^{\pi^t}(a_i) - v_i^{\pi^t} \right) \quad \text{(RMD)}$$
$$+ \mu \left( c_i(a_i) - \pi_i^t(a_i) \right).$$

The proof of this theorem is shown in Appendix D.1.

From here, we derive the relationship between the stationary point $\pi^\mu$ of (RMD) (i.e., the strategy profile that satisfies $\frac{d}{dt}\pi_i^\mu(a_i) = 0$ for all $i \in \{1, 2\}$ and $a_i \in A_i$) and the updated strategy profile $\pi^t$. Note that, from Lemma 3.3 in [Bauer et al., 2019], for any $\mu > 0$ there exists $\pi^\mu \in \prod_{i=1}^2 \Delta^\circ(A_i)$ such that $\pi^\mu$ is a stationary point of (RMD). Thus, $\pi^\mu$ is well-defined. We first derive the time derivative of the (sum of) Bregman divergence between $\pi^\mu$ and $\pi^t$:

**Theorem 5.2.** *Let $\pi^\mu \in \prod_{i=1}^2 \Delta(A_i)$ be a stationary point of (RMD). Then, $\pi^t$ updated by M-FTRL satisfies that:*

$$\frac{d}{dt} D_\psi(\pi^\mu, \pi^t)$$
$$= -\mu \sum_{i=1}^2 \sum_{a_i \in A_i} c_i(a_i) \left( \sqrt{\frac{\pi_i^t(a_i)}{\pi_i^\mu(a_i)}} - \sqrt{\frac{\pi_i^\mu(a_i)}{\pi_i^t(a_i)}} \right)^2.$$

*Furthermore, if the regularizer is entropy $\psi_i(p) = \sum_{a_i \in A_i} p(a_i) \ln p(a_i)$, then $\pi^t$ satisfies that:*

$$\frac{d}{dt} \mathrm{KL}(\pi^\mu, \pi^t) \leq -\mu\xi \mathrm{KL}(\pi^\mu, \pi^t),$$

*where $\xi = \min_{i \in \{1,2\}, a_i \in A_i} \frac{c_i(a_i)}{\pi_i^\mu(a_i)}$.*

The first statement implies that $\frac{d}{dt} D_\psi(\pi^\mu, \pi^t) = 0$ holds if and only if $\pi^t = \pi^\mu$, and $\forall \pi^t \neq \pi^\mu$, $\frac{d}{dt} D_\psi(\pi^\mu, \pi^t) < 0$. Thus, by Lyapunov arguments [Khalil, 2015], the Bregman divergence between $\pi^\mu$ and $\pi^t$ converges to 0, and then $\pi^t$ converges to $\pi^\mu$. Note that Theorem 5.2 holds for all stationary points of (RMD). This means that for a fixed $\mu$ and $(c_i)_{i=1}^2$, the stationary point is unique. From the second statement, we can show that exponential convergence rates can be achieved when using the entropy regularizer:

**Corollary 5.3.** *Assume that the regularizer is entropy $\psi_i(p) = \sum_{a_i \in A_i} p(a_i) \ln p(a_i)$. Then, M-FTRL's trajectory converges to a stationary point of (RMD) exponentially fast, i.e.,*

$$\mathrm{KL}(\pi^\mu, \pi^t) \leq \mathrm{KL}(\pi^\mu, \pi^0) \exp\left(-\mu\xi t\right).$$

Finally, combining this corollary and Lemma 3.5 in [Bauer et al., 2019], we can derive the exploitability bound of $\pi^t$.

**Theorem 5.4.** *Assume that the regularizer is entropy $\psi_i(p) = \sum_{a_i \in A_i} p(a_i) \ln p(a_i)$. Then, the exploitability for M-FTRL is bounded as:*

$$\mathrm{exploit}(\pi^t) \leq 2\mu$$
$$+ 2u_{\max}\sqrt{(\ln 2)\mathrm{KL}(\pi^\mu, \pi^0)} \exp\left(-\frac{\mu\xi}{2} t\right).$$

Theorem 5.4 means that $\pi^t$ converges to a $2\mu$-Nash equilibrium exponentially fast. The proof of the theorem is shown in Section 5.2.

## 5.1 PROOF SKETCH OF THEOREM 5.2

We sketch below the proof of Theorem 5.2. The complete proof and the associated lemmas are presented in Appendix D.2-D.4.

**Proof of the first part of Theorem 5.2.** First, we derive the time derivative of the Bregman divergence between $\pi \in \prod_{i=1}^2 \Delta(A_i)$ and $\pi^t$:

**Lemma 5.5.** *For any $\pi \in \prod_{i=1}^2 \Delta(A_i)$, $\pi^t$ updated by M-FTRL satisfies that:*

$$\frac{d}{dt} D_\psi(\pi, \pi^t)$$
$$= \sum_{i=1}^2 v_i^{\pi_i^t, \pi_{-i}} + 2\mu - \mu \sum_{i=1}^2 \sum_{a_i \in A_i} c_i(a_i) \frac{\pi_i(a_i)}{\pi_i^t(a_i)}.$$

The proof of Lemma 5.5 stems from the fact that $D_\psi(\pi, \pi^t) = \sum_{i=1}^2 \left( \max_{p \in \Delta(A_i)} \{ \langle z_i^t, p \rangle - \psi_i(p) \} - \langle z_i^t, \pi_i \rangle + \psi_i(\pi_i) \right)$. Next, we derive the relationship between the expected utilities $v^{\pi^\mu}$ and $v^{\pi_i', \pi_{-i}^\mu}$ for any $\pi_i' \in \Delta(A_i)$:

**Lemma 5.6.** *Let $\pi^\mu \in \prod_{i=1}^2 \Delta(A_i)$ be a stationary point of (RMD). Then, for any $i \in \{1, 2\}$ and $\pi_i' \in \Delta(A_i)$:*

$$v_i^{\pi_i', \pi_{-i}^\mu} = v_i^{\pi^\mu} + \mu - \mu \sum_{a_i \in A_i} c_i(a_i) \frac{\pi_i'(a_i)}{\pi_i^\mu(a_i)}.$$

This result can be shown by the fact that $\pi^\mu$ is the stationary point of (RMD), i.e., $\pi_i^\mu(a_i) \left( q_i^{\pi^\mu}(a_i) - v_i^{\pi^\mu} \right) + \mu \left( c_i(a_i) - \pi_i^\mu(a_i) \right) = 0$ for all $i \in \{1, 2\}$.

By combining Lemmas 5.5 and 5.6, we can obtain:

$$\frac{d}{dt} D_\psi(\pi^\mu, \pi^t)$$
$$= \sum_{i=1}^2 v_i^{\pi_i^t, \pi_{-i}^\mu} + 2\mu - \mu \sum_{i=1}^2 \sum_{a_i \in A_i} c_i(a_i) \frac{\pi_i^\mu(a_i)}{\pi_i^t(a_i)}$$
$$= \sum_{i=1}^2 v_i^{\pi^\mu} + 4\mu - \mu \sum_{i=1}^2 \sum_{a_i \in A_i} c_i(a_i) \left( \frac{\pi_i^t(a_i)}{\pi_i^\mu(a_i)} + \frac{\pi_i^\mu(a_i)}{\pi_i^t(a_i)} \right)$$
$$= 4\mu - \mu \sum_{i=1}^2 \sum_{a_i \in A_i} c_i(a_i) \left( \frac{\pi_i^t(a_i)}{\pi_i^\mu(a_i)} + \frac{\pi_i^\mu(a_i)}{\pi_i^t(a_i)} \right)$$
$$= -\mu \sum_{i=1}^2 \sum_{a_i \in A_i} c_i(a_i) \left( \sqrt{\frac{\pi_i^t(a_i)}{\pi_i^\mu(a_i)}} - \sqrt{\frac{\pi_i^\mu(a_i)}{\pi_i^t(a_i)}} \right)^2,$$

where the third equality follows from $\sum_{i=1}^2 v_i^{\pi^\mu} = 0$ by the definition of zero-sum games. This concludes the first statement of the theorem.

**Proof of the second part of Theorem 5.2.** Let us define $\xi_i = \min_{a_i \in A_i} \frac{c_i(a_i)}{\pi_i^\mu(a_i)}$. From the first part of the theorem,

we have:

$$\frac{d}{dt} D_\psi(\pi^\mu, \pi^t)$$

$$= -\mu \sum_{i=1}^{2} \sum_{a_i \in A_i} c_i(a_i) \left( \frac{\pi_i^t(a_i)}{\pi_i^\mu(a_i)} + \frac{\pi_i^\mu(a_i)}{\pi_i^t(a_i)} - 2 \right)$$

$$= -\mu \sum_{i=1}^{2} \sum_{a_i \in A_i} \frac{c_i(a_i)}{\pi_i^\mu(a_i)} \frac{(\pi_i^t(a_i) - \pi_i^\mu(a_i))^2}{\pi_i^t(a_i)}$$

$$\leq -\mu \sum_{i=1}^{2} \xi_i \sum_{a_i \in A_i} \frac{(\pi_i^t(a_i) - \pi_i^\mu(a_i))^2}{\pi_i^t(a_i)}$$

$$\leq -\mu \sum_{i=1}^{2} \xi_i \ln \left( 1 + \sum_{a_i \in A_i} \frac{(\pi_i^t(a_i) - \pi_i^\mu(a_i))^2}{\pi_i^t(a_i)} \right)$$

$$= -\mu \sum_{i=1}^{2} \xi_i \ln \left( \sum_{a_i \in A_i} \pi_i^\mu(a_i) \frac{\pi_i^\mu(a_i)}{\pi_i^t(a_i)} \right)$$

$$\leq -\mu \sum_{i=1}^{2} \xi_i \sum_{a_i \in A_i} \pi_i^\mu(a_i) \ln \left( \frac{\pi_i^\mu(a_i)}{\pi_i^t(a_i)} \right)$$

$$= -\mu \sum_{i=1}^{2} \xi_i \mathrm{KL}(\pi_i^\mu, \pi_i^t) \leq -\mu\xi \sum_{i=1}^{2} \mathrm{KL}(\pi_i^\mu, \pi_i^t), \quad (4)$$

where the second inequality follows from $x \geq \ln(1 + x)$ for all $x > 0$, and the third inequality follows from the concavity of the $\ln(\cdot)$ function and Jensen's inequality for concave functions. On the other hand, if $\psi_i(p) = \sum_{a_i \in A_i} p(a_i) \ln p(a_i)$, then $D_{\psi_i}(\pi^\mu, \pi^t) = \mathrm{KL}(\pi^\mu, \pi^t)$. From this fact and (4), we have:

$$\frac{d}{dt} \mathrm{KL}(\pi^\mu, \pi^t) \leq -\mu\xi \mathrm{KL}(\pi^\mu, \pi^t).$$

This concludes the second statement of the theorem. $\quad\square$

## 5.2 PROOF OF THEOREM 5.4

From the definition of exploitability, we have:

$$\mathrm{exploit}(\pi^t) = \sum_{i=1}^{2} \max_{\tilde{\pi}_i \in \Delta(A_i)} v_i^{\tilde{\pi}_i, \pi_{-i}^t}$$

$$= \sum_{i=1}^{2} \left( \max_{\tilde{\pi}_i \in \Delta(A_i)} v_i^{\tilde{\pi}_i, \pi_{-i}^\mu} \right.$$

$$\left. + \max_{\tilde{\pi}_i \in \Delta(A_i)} v_i^{\tilde{\pi}_i, \pi_{-i}^t} - \max_{\tilde{\pi}_i \in \Delta(A_i)} v_i^{\tilde{\pi}_i, \pi_{-i}^\mu} \right)$$

$$\leq \sum_{i=1}^{2} \left( \max_{\tilde{\pi}_i \in \Delta(A_i)} v_i^{\tilde{\pi}_i, \pi_{-i}^\mu} + \max_{\tilde{\pi}_i \in \Delta(A_i)} \left( v_i^{\tilde{\pi}_i, \pi_{-i}^t} - v_i^{\tilde{\pi}_i, \pi_{-i}^\mu} \right) \right)$$

$$\leq \sum_{i=1}^{2} \left( \max_{\tilde{\pi}_i \in \Delta(A_i)} v_i^{\tilde{\pi}_i, \pi_{-i}^\mu} \right.$$

$$\left. + \|\pi_i^\mu - \pi_i^t\|_1 \max_{\tilde{\pi}_{-i} \in \Delta(A_{-i})} \|q_i^{\pi_i^t, \tilde{\pi}_{-i}}\|_\infty \right)$$

$$\leq \sum_{i=1}^{2} \left( \max_{\tilde{\pi}_i \in \Delta(A_i)} v_i^{\tilde{\pi}_i, \pi_{-i}^\mu} + u_{\max} \sqrt{2(\ln 2) \mathrm{KL}(\pi_i^\mu, \pi_i^t)} \right), \tag{5}$$

where the second inequality follows from Hölder's inequality, and the last inequality follows from Lemma 11.6.1 in [Cover and Thomas, 2006].

From Lemma 3.5 of [Bauer et al., 2019], a stationary point $\pi^\mu$ of (RMD) satisfies that for all $i \in \{1, 2\}$ and $a_i \in A_i$, $q_i^{\pi^\mu}(a_i) - v_i^{\pi^\mu} \leq \mu$. Therefore, the term of $\max_{\tilde{\pi}_i \in \Delta(A_i)} v_i^{\tilde{\pi}_i, \pi_{-i}^\mu}$ can be bounded as:

$$\sum_{i=1}^{2} \max_{\tilde{\pi}_i \in \Delta(A_i)} v_i^{\tilde{\pi}_i, \pi_{-i}^\mu} = \sum_{i=1}^{2} \left( \max_{\tilde{\pi}_i \in \Delta(A_i)} v_i^{\tilde{\pi}_i, \pi_{-i}^\mu} - v_i^{\pi^\mu} \right)$$

$$= \sum_{i=1}^{2} \left( \max_{a_i \in A_i} q_i^{\pi^\mu}(a_i) - v_i^{\pi^\mu} \right) \leq 2\mu, \tag{6}$$

where the second equality follows from $\sum_{i=1}^{2} v_i^{\pi^\mu} = 0$ by the definition of zero-sum games. By combining (5), (6), and Corollary 5.3, we have:

$$\mathrm{exploit}(\pi^t) \leq 2\mu + u_{\max} \sum_{i=1}^{2} \sqrt{2(\ln 2) \mathrm{KL}(\pi_i^\mu, \pi_i^t)}$$

$$\leq 2\mu + u_{\max} \sqrt{2(\ln 2)} \sqrt{2 \sum_{i=1}^{2} \mathrm{KL}(\pi_i^\mu, \pi_i^t)}$$

$$\leq 2\mu + 2\sqrt{\ln 2} u_{\max} \sqrt{\mathrm{KL}(\pi^\mu, \pi^0) \exp(-\mu\xi t)}$$

$$= 2\mu + 2u_{\max} \sqrt{(\ln 2) \mathrm{KL}(\pi^\mu, \pi^0)} \exp\left( -\frac{\mu\xi}{2} t \right),$$

where the second inequality follows from $\sqrt{a} + \sqrt{b} \leq \sqrt{2(a + b)}$ for $a, b > 0$. This concludes the statement. $\quad\square$

## 6 EXPERIMENTS

In this section, we empirically evaluate M-FTRL. We compare its performance to those of FTRL and O-FTRL.

We conduct experiments on the following games: biased rock-paper-scissors (BRPS), a normal-form game with multiple Nash equilibria (M-Eq), and random utility games. BRPS and M-Eq have the following utility matrix, respectively:

Table 1: Biased RPS utilities

|   | R | P | S |
|---|---|---|---|
| R | 0 | −0.1 | 0.3 |
| P | 0.1 | 0 | −0.1 |
| S | −0.3 | 0.1 | 0 |

Table 2: M-Eq utilities

|   | $y_1$ | $y_2$ |
|---|---|---|
| $x_1$ | 0.1 | −0.2 |
| $x_2$ | −0.4 | 0.3 |
| $x_3$ | −1 | 0.9 |

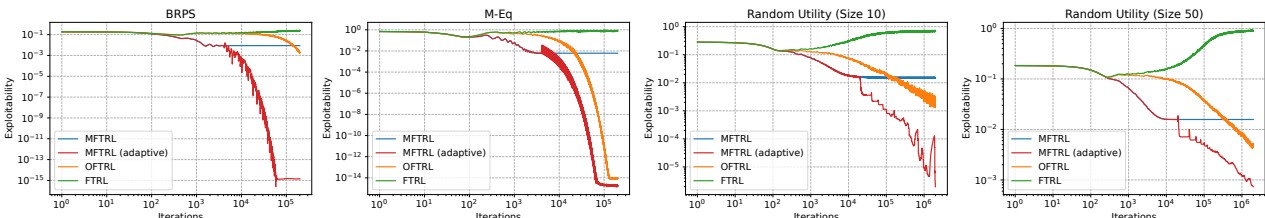

Figure 2: Exploitability of $\pi^t$ for M-FTRL, FTRL, and O-FTRL under full-information feedback.

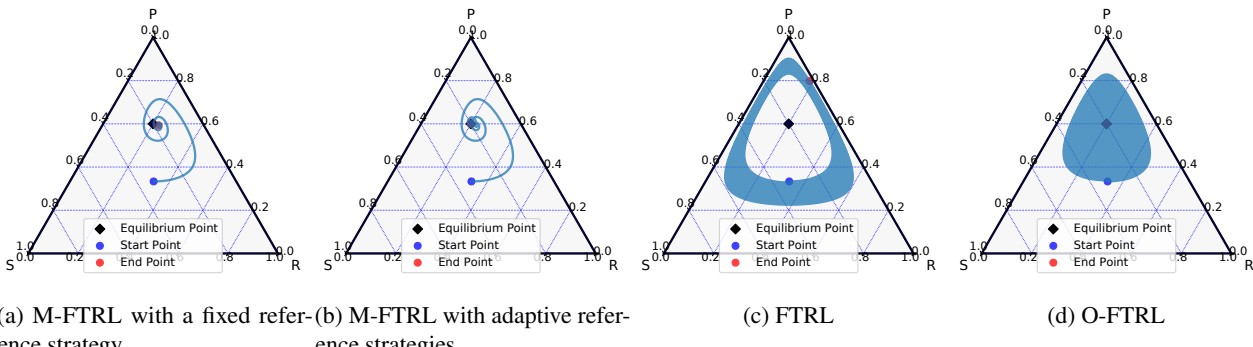

(a) M-FTRL with a fixed reference strategy
(b) M-FTRL with adaptive reference strategies
(c) FTRL
(d) O-FTRL

Figure 3: Trajectories of $\pi^t$ for M-FTRL, FTRL and O-FTRL in BRPS under full-information feedback. We set the initial strategy profile to $\pi_i^0 = \frac{1}{|A_i|}$ for $i \in \{1, 2\}$. The black point represents the equilibrium strategy. The blue/red points represent the initial/final points, respectively.

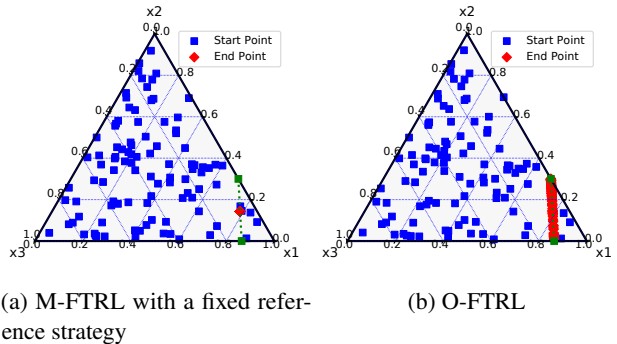

(a) M-FTRL with a fixed reference strategy
(b) O-FTRL

Figure 4: Initial strategies and final strategies for player 1 in 100 instances (M-Eq under full-information feedback). The green dashed line represents the set of equilibrium strategies for player 1. The blue/red points represent the initial/final points, respectively.

The set of Nash equilibria in M-Eq is given by:

$$\Pi_1^* = \left\{ x \in \Delta^3 \mid x_2 = -\frac{22}{12}x_1 + \frac{19}{12}; \; x_3 = \frac{10}{12}x_1 - \frac{7}{12} \right\},$$

$$\Pi_2^* = \left\{ \left( \frac{1}{2}, \frac{1}{2} \right) \right\}.$$

For random utility games, we generate each component in a utility matrix uniformly at random in $[0, 1]$. We consider random utility games with action sizes $|A_1| = |A_2| = 10$ and $|A_1| = |A_2| = 50$. For each game, we average the results

for 100 instances. We generate the initial strategy profile $\pi^0$ uniformly at random in $\prod_{i=1}^2 \Delta^\circ(A_i)$ for each instance. We use the entropy regularizer $\psi_i(p) = \sum_{a_i \in A_i} p(a_i) \ln p(a_i)$ in all experiments.

### 6.1 FULL-INFORMATION FEEDBACK

First, we provide the results under full-information feedback. In these experiments, we analyze the performance of M-FTRL with a fixed reference strategy $c_i = \left( \frac{1}{|A_i|} \right)_{a_i \in A_i}$ and one with adaptive reference strategies (Algorithm 1). We set the learning rate to $\eta = 10^{-1}$ for all algorithms, and set the mutation parameter to $\mu = 10^{-2}$ for M-FTRL. For M-FTRL with adaptive reference strategies, we set $N = 4,000$ in BRPS and M-Eq, and $N = 20,000$ in the random utility games.

Figure 2 shows the average exploitability of $\pi^t$ updated by each algorithm. We find that the exploitability of M-FTRL converges to a constant value faster than FTRL and O-FTRL. Furthermore, by adapting the reference strategy, the exploitability of M-FTRL's strategy profile quickly converges to 0. We provide additional experimental results with varying mutation parameters $\mu \in \{10^{-3}, 5 \times 10^{-3}, 10^{-2}, 10^{-1}, 1\}$ in Appendix B.

Next, we compare the trajectories of strategies updated by each algorithm. Figure 3 shows the trajectories of $\pi^t$ updated by each algorithm from an instance of RBPS. Note

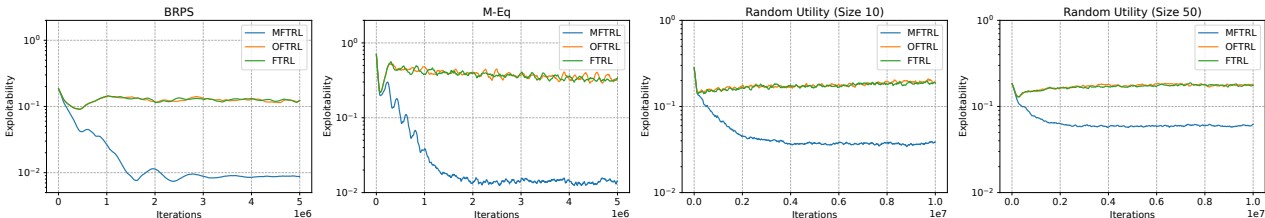

Figure 5: Exploitability of $\pi^t$ for M-FTRL, FTRL, and O-FTRL under bandit feedback.

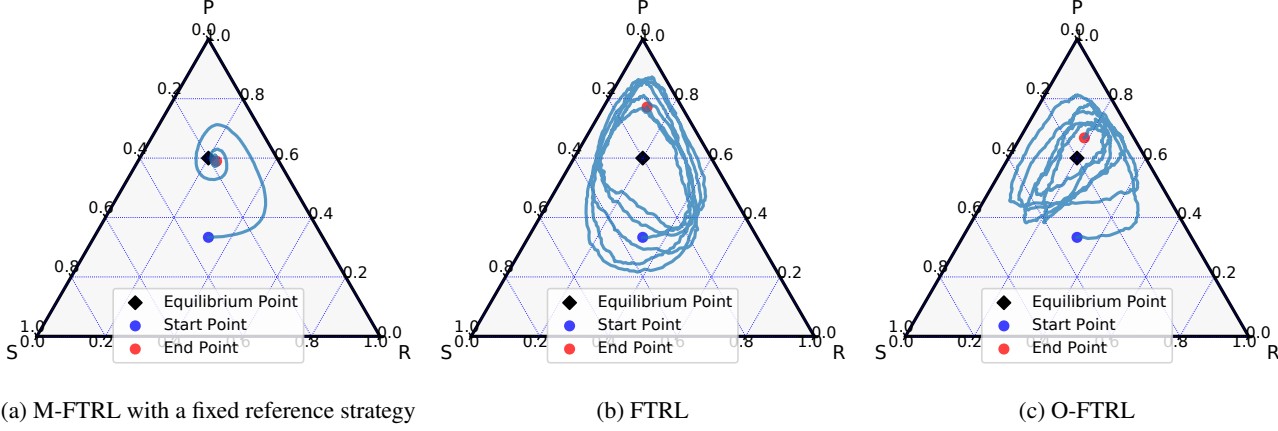

(a) M-FTRL with a fixed reference strategy

(b) FTRL

(c) O-FTRL

Figure 6: Trajectories of $\pi^t$ for M-FTRL, FTRL and O-FTRL in BRPS under bandit feedback. We set the initial strategy profile to $\pi_i^0 = \frac{1}{|A_i|}$ for $i \in \{1, 2\}$. The black point represents the equilibrium strategy. The blue/red points represent the initial/final points, respectively.

that in this figure, we set the initial strategy to $\pi_i^0 = \frac{1}{|A_i|}$ for $i \in \{1, 2\}$. We can observe that FTRL's strategies cycle around the Nash equilibrium strategy, and O-FTRL's strategies gradually approach the Nash equilibrium strategy. Unlike these methods, M-FTRL's strategies quickly approach the stationary point. Figure 4 shows the initial strategies and final strategies for player 1 in M-Eq. We find that M-FTRL's strategy profile converges to a unique stationary point regardless of the setting of the initial point, while O-FTRL's strategy profile converges to a different Nash equilibrium for each instance. This result highlights the uniqueness property of the stationary point from Theorem 5.2.

### 6.2 BANDIT FEEDBACK

Next, we provide the results under bandit feedback. We set the learning rate to $\eta = 10^{-4}$ for all algorithms, and set the mutation parameter to $\mu = 10^{-2}$ for M-FTRL. In the bandit feedback experiments, we focus on the performance of M-FTRL with a fixed reference strategy $c_i = \frac{1}{|A_i|}$. In FTRL and O-FTRL algorithms, we use the unbiased estimator by [Lattimore and Szepesvári, 2020] as the estimator of $q_i^{\pi^t}$ so that the estimator takes values in $(-\infty, u_{\max}]$ for computational stability. We provide further details on the estimator in Appendix A. Note that M-FTRL does not need this estimator, but it is sufficient to use the importance-

weighted estimator in (2).

Figure 5 shows the average exploitability of $\pi^t$ updated by each algorithm, and Figure 6 shows the trajectories of $\pi^t$ updated by each algorithm from an instance of RBPS. We can see that unlike the experimental results under full-information feedback, O-FTRL's trajectory does not converge to a Nash equilibrium. On the other hand, M-FTRL's trajectory converges near a stationary point. These results suggest that M-FTRL has the last-iterate convergence property even under bandit feedback.

## 7 CONCLUSION

In this study, we proposed M-FTRL, a simple FTRL algorithm that incorporates mutation for last-iterate convergence to a stationary point. We proved that the M-FTRL dynamics induced by the entropy regularizer is equivalent to RMD. Besides, we showed that the trajectory of M-FTRL with general regularization functions converges to a stationary point of the RMD. The numerical simulation reveals that M-FTRL outperforms the state-of-the-art FTRL and O-FTRL in a variety of two-player zero-sum games. In future studies, we will extend M-FTRL algorithm and provide a theoretical analysis to more complex games, such as extensive-form games and Markov games.

**Acknowledgements**

Atsushi Iwasaki was supported by JSPS KAKENHI Grant Numbers JP21H04890 and JP20K20752.

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
