# OpenReview forum: "Mutation-Driven Follow the Regularized Leader for Last-Iterate Convergence in Zero-Sum Games"
_auai.org/UAI/2022/Conference — UAI 2022 Poster_

### Official Review · Reviewer_Xth2 · 2022-03-26

**Q2(1) Originality/Novelty:** 3
**Q2(2) Significance/Impact:** 3
**Q2(3) Correctness/Technical Quality:** 3
**Q2(6) Clarity Of Writing:** 3
**Q6 Overall Score:** 6
**Q8 Confidence In Your Score:** 3

**Q1 Summary And Contributions:**

This work investigates the last-iterate convergence in two-player zero-sum games. Previous methods such as optimistic FTRL can converge. This work offers an alternative method called mutation-driven FTRL which also enjoys last-iterate convergence. The main contributions are this work is theoretical. The empirical studies also validate the effectiveness of the proposed method.

**Q2 Assessment Of The Paper:**

More detailed information regarding each of these aspects is given below:

**Q2(4) Quality Of Experiments (Optional):**

3: Good: The experimental evaluation is adequate, and the results convincingly support the main claims.

**Q2(5) Reproducibility:**

2: Fair: Key resources (e.g., proofs, code, data) are unavailable but key details (e.g., proof sketches, experimental setup) are sufficiently well-described for an expert to confidently reproduce the main results.

**Q3 Main Strengths:**

1. The problem is well-motivated and studied. The two-player zero-sum game is a basic model in game theory and the last-iterate convergence is also a property as important as average-iterate convergence.
2. The theoretical contributions are sufficient: 1) M-FTRL recovers a previous dynamics called RMD; 2) for general regularization functions, the strategy trajectory of M-FTRL converges to a stationary point of the RMD; 3) first to provide the exponential convergence result for RMD.
3. The experiments are convincing. The authors compare the proposed methods with FTRL and optimistic FTRL, showing the effectiveness of the proposed method in several cases and matching the theoretical findings.

**Q4 Main Weakness:**

Not much.

**Q5 Detailed Comments To The Authors:**

A little suggestion of the experiment part: in Figure 2, some parts of the first two subfigures are clipped.

**Q7 Justification For Your Score:**

See Q3.

**Q9 Complying With Reviewing Instructions:**

1: Yes.

---

### Official Review · Reviewer_yqW3 · 2022-04-07

**Q2(1) Originality/Novelty:** 3
**Q2(2) Significance/Impact:** 4
**Q2(3) Correctness/Technical Quality:** 4
**Q2(6) Clarity Of Writing:** 3
**Q6 Overall Score:** 7
**Q8 Confidence In Your Score:** 3

**Q1 Summary And Contributions:**

The paper studies zero-sum games, for which a variant of the popular Follow-the-regularized-leader (FTRL) is suggested. The variant involves a perturbation of the FTRL action probabilities inspired by mutation dynamics. The time-continuous counterpart of the algorithm is analyzed regarding its convergence properties, while the actual discrete approach is investigated in an empirical study on three different zero-sum games.

**Q2 Assessment Of The Paper:**

More detailed information regarding each of these aspects is given below:

**Q2(4) Quality Of Experiments (Optional):**

3: Good: The experimental evaluation is adequate, and the results convincingly support the main claims.

**Q2(5) Reproducibility:**

2: Fair: Key resources (e.g., proofs, code, data) are unavailable but key details (e.g., proof sketches, experimental setup) are sufficiently well-described for an expert to confidently reproduce the main results.

**Q3 Main Strengths:**

- Zero-sum games are relevant in many fields of research, so the paper considers a certainly relevant topic with a broad range of applications.
- Except for minor things, the paper is well written and has a reasonable structure. Also I think it provides a good overview of the relevant literature (although I am not an expert on this field).

-  The suggested variant of the popular FTRL algorithm is non-trivial. Also, the link of its time-continuous counterpart to replicator dynamics is quite interesting (and presumably how the discrete-time variant has been derived).

- The  theoretical results as well as the empirical properties of the algorithm are appealing. The proofs of the theoretical results seem to be sound and are written in a way that they are really easy to follow.


**Q4 Main Weakness:**

- The theoretical analysis is ‘’only’’ for the continuous-time counterpart of the suggested algorithm, which, however, is not so trivial on its own. Moreover, the bandit-feedback case is not considered theoretically, although it is studied in the experiments.


- There is no justification for the choices of the hyperparameters used in the experiments, i.e., the mutation rate and the frequency of copying the profile probability for the mutation probability. A short notice such as ‘’we found that a mutation rate in the range [a,b] works well’’ would be good in order to give a hint for practically interested readers.

- As a non-expert in this field, I found the second paragraph of the introduction a bit difficult to read (in the 1st iteration), as concepts such as ‘’last-iterate-convergence’’ or ‘’last strategy’’, where assumed to be known. Perhaps a short intuitive description of these notions would help in this regard.

**Q5 Detailed Comments To The Authors:**

*Update of* $q_i^{\pi_t}$: A clearer formulation of how the feedback received (i.e., the realized utility $u_i(a_1^t, a_2^t)$) is used to update the selection mechanism would be desirable.

*Notation*: In general the notation is well thought out, but I still have some suggestions to improve it:

- $\Delta(A_i)$ has never been defined (although one can guess that it is the simplex). Introducing it rigorously would also help to understand the meaning of $p(a_i)$ for instance.

- In the proof section, you use the shorthand notation $A$ (which is $A_i \times A_{-i}$) and $u_i(a)$ for $a=(a_i,a_{-i})$ without introducing it either.

- As the paper uses quite some notation it would be perhaps sensible to provide a list of frequently used symbols in the appendix.


**Other minor things:**

- Stationary points: I would suggest to recall the mathematical definition of a stationary point directly after Theorem 5.1.

- The reference to Shalev-Shwartz is not correct, as the article is written only by one author.

- p.2 (left column): `` … no-regret algorithms [which] enjoy … ‘’

- Eq.(1): there should be an $s$ in the superscript of $q$ instead of a $t$. And a bit below I think it should be Figure 1 (c) or (d) instead of Figure 1 a.

-   End of proof of Lemma 5.5: “… where the third equality … and [the fourth equality] from …”. Then it should be $\pi_i$ (instead of $\pi_i^t$) on the left-hand side of the last line. Also, it would do no harm to remark that the fifth equality is by definition of the zero-sum game (for sake of completeness).

# Post Rebuttal

After reading the other reviews and the author's response, I am even more positive about the paper.


**Q7 Justification For Your Score:**

Although the paper has some weaknesses, I think it is overall an interesting contribution to the existing literature. The proposed tweak of the FTRL algorithm is interesting and more importantly sensible, as the theoretical analysis of the time-continuous counterpart shows, as well as the results of the empirical study. The only weak spot is that no theoretical analysis is provided for the actual discrete version of the suggested algorithm and for the bandit feedback case.

**Q9 Complying With Reviewing Instructions:**

1: Yes.

---

### Official Review · Reviewer_nA7M · 2022-04-14

**Q2(1) Originality/Novelty:** 2
**Q2(2) Significance/Impact:** 3
**Q2(3) Correctness/Technical Quality:** 3
**Q2(6) Clarity Of Writing:** 3
**Q6 Overall Score:** 6
**Q8 Confidence In Your Score:** 2

**Q1 Summary And Contributions:**

This paper considers the problem of learning an equilibrium in two-player zero-sum games.  The authors propose a mutation driven Follow The Regularized Leader algorithm (M-FTRL) whose last strategy converges. A discrete-time version and a continuous-time version of the proposed approach are provided. It is proven that the trajectory of M-FTRL converges to a stationary point. The provided simulation  results show that M-FTRL performs well under full-information feedback and bandit feedback.

**Q2 Assessment Of The Paper:**

More detailed information regarding each of these aspects is given below:

**Q2(4) Quality Of Experiments (Optional):**

3: Good: The experimental evaluation is adequate, and the results convincingly support the main claims.

**Q2(5) Reproducibility:**

2: Fair: Key resources (e.g., proofs, code, data) are unavailable but key details (e.g., proof sketches, experimental setup) are sufficiently well-described for an expert to confidently reproduce the main results.

**Q3 Main Strengths:**

1. The mathematical treatment of the considered problem is through.

2. The paper is very well-written and the main ideas as well as the mathematical details are generally presented well.

**Q4 Main Weakness:**

The code is not available.

**Q5 Detailed Comments To The Authors:**

Please consider adding more details about the experimental setup to the example given in the beginning of section 4. At this point in the paper, M-FTRL is yet to be introduced, so it is hard to grasp the interpretation of the mutations given in Fig 1b, 1c an 1d.

**Q7 Justification For Your Score:**

The proposed solution seems suitable and the provided results are strong however I am not sure about the novelty of the work.

**Q9 Complying With Reviewing Instructions:**

1: Yes.

---

### Decision · Program_Chairs · 2022-05-15

**Decision:**

Accept (Poster)

**Comment:**

Meta Review: The reviewers are in agreement that this is a strong contribution.